# Skyrmions, quantum Hall droplets, and one current to rule them all

Avner Karasik[*]

Department of Applied Mathematics and Theoretical Physics,
University of Cambridge, Cambridge, CB3 0WA, UK

* avnerkar@gmail.com

## Abstract

We introduce a novel Skyrme-like conserved current in the effective theory of pions and vector mesons based on the idea of hidden local symmetry. The associated charge is equivalent to the skyrmion charge for any smooth configuration. In addition, there exist singular configurations that can be identified as $N_f = 1$ baryons charged under the new symmetry. Under this identification, the vector mesons play the role of the Chern-Simons vector fields living on the quantum Hall droplet that forms the $N_f = 1$ baryon. We propose that this current is the correct effective expression for the baryon current at low energies. This proposal gives a unified picture for the two types of baryons and allows them to continuously transform one to the other in a natural way. In addition, Chern-Simons dualities on the droplet can be interpreted as a result of Seiberg-like duality between gluons and vector mesons.



# 1 Introduction

In this paper, we study and compare the low energy description of baryons in $N_f \geq 2$ QCD, also known as skyrmions [1–4] with the low energy description of baryons in $N_f = 1$ QCD, recently constructed by Komargodski in [5]. These two objects look very different at low energies. Skyrmions enjoy a complete description as solitons in the effective theory of pions. They are topologically stable finite energy configurations, thanks to $\Pi_3(SU(N_f)) = \mathbb{Z}$ for every $N_f \geq 2$. $N_f = 1$ baryons, on the other hand, cannot be described completely using effective (mesonic) degrees of freedom. In [5] they were constructed using the $\eta'$ field as a smooth configuration everywhere in space except for a singular ring. $\eta'$ winds around the ring which implies that the vacuum expectation value (VEV) of the chiral condensate must vanish on the ring. One can argue that there should be a $U(1)_N$ Chern-Simons (CS) theory living on the $\eta' = \pi$ disc bounded by the ring. As in the quantum Hall effect, quantization of the CS theory with Dirichlet boundary conditions leads to a chiral boson living on the boundary. The combination of the $\eta'$ winding around the ring, and the chiral boson winding along the ring forms a stable soliton which can be identified as the $N_f = 1$ baryon.

The main goal of this paper is to give a unified description of the two different types of baryons. In particular, we would like to claim that the correct low energy description of the baryon current is

$$
H^\mu = \frac{1}{24\pi^2} \epsilon^{\mu\nu\rho\sigma} \text{tr} \left[ 2\partial_\nu \xi \xi^\dagger \partial_\rho \xi \xi^\dagger \partial_\sigma \xi \xi^\dagger + 3i V_\nu (\partial_\rho \xi \partial_\sigma \xi^\dagger - \partial_\rho \xi^\dagger \partial_\sigma \xi) \right.
$$
$$
\left. + 3i \partial_\nu V_\rho (\partial_\sigma \xi \xi^\dagger - \partial_\sigma \xi^\dagger \xi) \right] , \tag{1}
$$

where $V_\mu$ are the $U(N_f)$ vector mesons and $\xi \in U(N_f)$ is roughly the square root of the unitary pion+$\eta'$ matrix $\xi^2 = U \in U(N_f)$. The derivation of this current is based on the idea of hidden local symmetry [6, 7]. The charge computed using $H^\mu$ is equivalent to the usual skyrmion charge for any smooth configuration. In addition, there exists non-smooth configurations charged under $H^\mu$ and not charged under the usual skyrmion current. The $N_f = 1$ baryon is exactly such a configuration. More precisely, for $N_f = 1$ QCD,

$$
H^\mu_{(N_f=1)} = -\frac{1}{8\pi^2} \epsilon^{\mu\nu\rho\sigma} \partial_\nu \omega_\rho \partial_\sigma \eta' , \tag{2}
$$

where $\omega_\mu = tr(V_\mu)$ is the $U(1)$ vector meson.[1] This expression is equivalent to the charge of the $N_f = 1$ baryon if we identify the $\omega_\mu$ vector meson with the $U(1)_N$ CS vector field mentioned above. We will give some evidence and discuss some of the consequences of this identification.

The outline of the paper is as follows. In section 2 we will review some basic facts about $N_f \geq 2$ skyrmions and $N_f = 1$ quantum Hall droplets. In section 3 we will show how some of the features of the $N_f = 1$ baryon emerge when continuously flowing from $N_f = 2$ to $N_f = 1$ by taking one of the quarks' masses to be very large. Section 4 contains the main results of the paper. We will start by adding the vector mesons to the low energy effective theory and reviewing the concept of hidden local symmetry. Later, we will present the current $H^\mu$ and its relation to the two types of baryons. In section 5 we will discuss the proposal of identifying the $\omega_\mu$ vector meson as the CS vector field on the $\eta' = \pi$ domain wall, including interesting relations to 3d CS dualities and (non-supersymmetric) Seiberg dualities. In section 6 we will discuss some additional details and some open problems related to the edge modes living on the ring.

---

[1]See also equation (64) in [28] for a similar expression for the current.

## 2 Background

### 2.1 $N_f \geq 2$ Skyrmions: review

In this section we will review some of the basic facts about skyrmions. Our starting point is $SU(N)$ QCD with $N_f \geq 2$ massless Dirac fermions. The theory enjoys the global symmetry[2] of $SU(N_f)_L \times SU(N_f)_R \times U(1)_B$. In addition, the QCD Lagrangian enjoys the axial symmetry $U(1)_A$ which is broken by non-perturbative effects. However, in the large $N$ limit, the symmetry is restored and $U(1)_A$ becomes an exact symmetry of the theory. For $N_f$ not too large (below the conformal window), the theory is confining at low energies, and the symmetries are spontaneously broken by the chiral condensate

$$SU(N_f)_L \times SU(N_f)_R \times U(1)_B \times U(1)_A \to SU(N_f)_V \times U(1)_B , \tag{3}$$

where $SU(N_f)_V$ is the diagonal subgroup of $SU(N_f)_L \times SU(N_f)_R$ leaving the chiral condensate invariant. We also included $U(1)_A$ even though it is at best only an approximate symmetry for any finite $N$. The low energy effective theory can be described using Goldstone theorem by a non-linear sigma model, parametrized by $U(x) \in U(N_f)$. The global symmetries act on $U$ as

$$U \to e^{i\alpha} V_L^\dagger U V_R , \ V_{L,R} \in SU(N_f)_{L,R} , \ e^{i\alpha} \in U(1)_A . \tag{4}$$

Indeed, the vacuum $U = 1$ breaks the symmetries as described in (3). $U(1)_B$ on the other hand doesn't act on $U$. From the microscopic point of view, the only gauge invariant operators charged under $U(1)_B$ are the baryons

$$B^{i_1...i_N} = \epsilon^{a_1...a_N} \psi_{a_1}^{i_1}...\psi_{a_N}^{i_N} , \tag{5}$$

where $a_{1,...,N}$ are color indices and $i_{1,..,N}$ are flavor indices. A surprising fact about baryons is that even though we wrote an effective theory only for the massless Nambu-Goldstone (NG) modes and thrown away all the rest, baryons still appear as solitons, famously known as skyrmions. For the rest of the section we will restrict $U(x) \in SU(N_f)$ since the $U(1)$ plays no role in the construction of skyrmions. The effectve theory is described by the chiral Lagrangian

$$\mathscr{L} = \frac{F_\pi^2}{4} \text{tr}(\partial_\mu U^\dagger \partial^\mu U) + ... . \tag{6}$$

The ... includes higher derivatives terms and for $N_f \geq 3$ also the Wess-Zumino term. For any finite energy configuration, the fields must go to their vacuum at infinity $\lim_{r\to\infty} U(x) = 1$. Finite energy configurations are maps from $S^3$ to $SU(N_f)$ which are classified by

$$\Pi_3(SU(N_f)) = \mathbb{Z} \ \forall \ N_f \geq 2 , \tag{7}$$

which allows the existence of stable solitons. The associated topological current is the skyrmion current

$$B^\mu = \frac{1}{24\pi^2} \epsilon^{\mu\nu\rho\sigma} \text{tr}(U^\dagger \partial_\nu U U^\dagger \partial_\rho U U^\dagger \partial_\sigma U) , \tag{8}$$

which is identically conserved $\partial_\mu B^\mu = 0$ and the associated charge is $B = \int d^3x B^t \in \mathbb{Z}$. We will focus now on the simple case of $N_f = 2$. A convenient parametrization of $U \in SU(2)$ is

$$U = \sigma + i\tau_a \pi_a , \ \sigma^2 + \pi_a^2 = 1 , \tag{9}$$

---

[2]We consider here only the continuous symmetries. See for example [8] for a recent discussion about the discrete factors.

where $\tau_a$ are the Pauli matrices. An example for a charged configuration is the hedgehog ansatz

$$U = \cos(f(r)) + \frac{i\sin(f(r))x_a\tau_a}{r} \ . \tag{10}$$

The condition $U(r \to \infty) = 1$ can be satisfied by taking $f(r \to \infty) = 0$ without loss of generality. Demanding that $U$ has a well defined limit at the origin requires $\sin(f(r=0))$ to vanish, which implies $f(0) = \pi K$ for some integer $K$. It is a straight forward exercise to show that for this configuration

$$B = K \ . \tag{11}$$

There are many pieces of evidence and consistency checks that the skyrmions indeed should be identified with baryons, and that the topological symmetry (8) is the low energy description of $U(1)_B$. These include the spin, coupling to chiral gauge fields, large $N$ and many more (see for example [4, 9–17]). For any $N_f > 2$, the story works basically the same by choosing an $SU(2) \subset SU(N_f)$ and embedding the hedgehog solution in this subgroup. For $N_f = 1$ the story is more complicated. For $N_f = 1$ the theory is gapped as there are no NG bosons. Any effective description of baryons, if it exists, must include other degrees of freedom.

## 2.2 $N_f = 1$ quantum Hall droplet: review

In this section we will review the recent work by Komargodski [5] in which he constructed a soliton that can identified with the $N_f = 1$ baryon. From the microscopic point of view, $N_f = 1$ baryons can be written as

$$\epsilon^{a_1...a_N}\psi_{a_1}...\psi_{a_N} \ . \tag{12}$$

Due to the anti-symmetrization over color indices, and the fermionic nature of $\psi$, the spin indices must be symmetrized over to get something which is not identically zero. Therefore, there exists only one type of $N_f = 1$ baryon and its spin is $\frac{N}{2}$. The low energy effective theory is gapped. However, as mentioned above, in the large $N$ limit $U(1)_A$ becomes an exact symmetry, and its breaking leads to a NG boson known as the $\eta'$. $\eta'$ is a periodic scalar $\eta' \simeq \eta' + 2\pi$. The effective Lagrangian including the leading $\frac{1}{N}$ correction is given by

$$\mathscr{L}_{\eta'} = \frac{F_\pi^2}{2}(\partial\eta')^2 - \frac{F_\pi^2 M_{\eta'}^2}{2}\min_{k\in\mathbb{Z}}(\eta' + 2\pi k)^2 \ , \ M_{\eta'}^2 \sim O\left(N^{-1}\right) \ . \tag{13}$$

The potential term is locally quadratic but has a cusp whenever $\eta' = \pi$ mod $2\pi$. For small fluctuations around the vacuum $\eta'_{vac} = 0$ it simply looks like a mass term, but when global effects that include non-trivial winding of $\eta'$ are present, the cusp plays an important role. The physical interpretation of the cusp is that when $\eta'$ crosses $\pi$, heavy fields jump from one vacuum to the other. [18] This cusp is closely related to the first order phase transition in pure Yang-Mills theory (YM) when $\theta = \pi$. [18–22] The simplest way to see this is to notice that due to the ABJ anomaly, axial transformations lock shifts of $\eta'$ by a constant with shifts of $\theta$ by the same constant $\eta' \to \eta' + \alpha \Leftrightarrow \theta \to \theta + \alpha$. For $\theta = \pi$ YM, the domain wall connecting the two vacua must carry a TFT on its worldvolume. More precisely, YM at $\theta = \pi$ has a mixed 't-Hooft anomaly between time reversal and the $\mathbb{Z}_N$ 1-form symmetry. The domain wall connects two vacua related by the action of time reversal, which implies that the theory on the domain wall must carry an anomalous $\mathbb{Z}_N$ 1-form symmetry. The desired anomaly is matched by $U(1)_N$ Chern-Simons (CS) theory.[3] It is natural to conjecture that also for $N_f = 1$ QCD, a configuration that interpolates between $\eta' = 0$ to $\eta' = 2\pi$ carries a $U(1)_N$ CS theory on the sheet $\eta' = \pi$.

---

[3]There is also a dual description in terms of an $SU(N)_{-1}$ CS theory, but for us the first description will be more convenient.

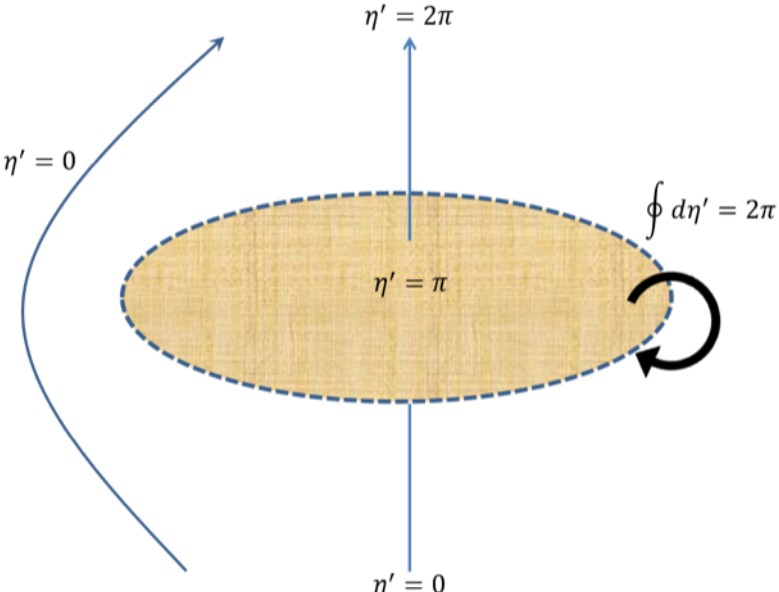

Figure 1: The $N_f = 1$ baryon of [5]. In the figure, the pancake is schematically the $\eta' = \pi$ sheet where the CS theory lives. For any closed trajectory that goes through the pancake, $\eta'$ winds from 0 to $2\pi$.

The theory (13) enjoys a topological $U(1)$ 2-form symmetry, associated with the current

$$J_{\mu\nu\rho} = \frac{1}{2\pi}\epsilon_{\mu\nu\rho\sigma}\partial^\sigma\eta' \, . \tag{14}$$

Charged objects under this symmetry are infinitely extended sheets that interpolate from $\eta' = 0$ on one side to $\eta' = 2\pi$ on the other [23, 24]. As an example, consider the configuration

$$\eta' = f(z) \, , \ \lim_{z\to-\infty} f(z) = 0 \, , \ \lim_{z\to\infty} f(z) = 2\pi \, . \tag{15}$$

Indeed, the configuration satisfies[4]

$$Q = \int dz J_{txy} = 1 \, . \tag{16}$$

One problem with these sheets is that while their tension is finite, their mass $\sim \int dx dy$ diverges. One cannot construct finite energy configurations charged under this symmetry in 3+1 dimensions. Instead, we can consider finite sheets of the following schematic form. To get finite energy, we must demand that $\lim_{r\to\infty} \eta'(\vec{r}) = 0 \mod 2\pi$. In addition, we will try to impose that $\eta'(x = y = 0, z) = f(z)$ as before, with $f(0) = \pi$. These two demands cannot live together without having singularities somewhere in space. The minimal singularity that must exist is of the form of a ring, surrounding the $\eta' = \pi$ sheet. The configuration is illustrated in figure 1 where it can be seen that $\eta'$ must wind from 0 to $2\pi$ as we go around the ring.

A key question is what happens on the ring. We can expect that as we go closer and closer to the ring, the chiral condensate goes to zero until it vanishes exactly on the ring. The physics on the ring is therefore beyond the scope of the low energy effective theory (13). A progress

---

[4]Notice that because this is a 2-form symmetry, the charge is codimension 3. See [25] for more details.

can still be made if we think of the ring as the boundary of the CS theory living on the $\eta' = \pi$ sheet. Consider the $U(1)_N$ CS theory on a disc of radius 1,

$$\mathscr{L}_{CS} = \frac{N}{4\pi}\epsilon^{\mu\nu\rho}a_\mu\partial_\nu a_\rho \,. \tag{17}$$

Under a general variation $a_\mu \to a_\mu + \delta a_\mu$, the action transforms as

$$\delta S_{CS} = \frac{N}{2\pi}\int d^3x\,\epsilon^{\mu\nu\rho}\partial_\mu a_\nu \delta a_\rho + \frac{N}{4\pi}\int d\phi\,dt(a_\phi \delta a_t - a_t \delta a_\phi)\,, \tag{18}$$

where $\phi$ is the angular coordinate on the boundary. For the specific choice of gauge variations $a_\mu \to a_\mu + \partial_\mu \lambda$, the transformation of the action is

$$\delta S_{CS} = \frac{N}{4\pi}\int d\phi\,dt\,\lambda(\partial_\phi a_t - \partial_t a_\phi)\,. \tag{19}$$

The theory can be quantized as follows. In order to have a well defined variational principle we impose Dirichlet boundary conditions, $a_t = v a_\phi$ such that the boundary term in (18) vanishes identically. In addition, Lorentz invariance leads us to choose $v = 1$. See footnote (13) of [5] for more details on this point. Gauge invariance then implies that on the boundary,

$$(\partial_\phi - \partial_t)a_\phi = 0 \Rightarrow a_\phi = a_\phi(\phi + t)\,. \tag{20}$$

The bulk term in (18) gives the equations of motion (EOM), $F_{\mu\nu} = 0$. The EOM are solved by having $a_\mu = \partial_\mu \lambda$ everywhere. However, $a_\mu$ can still be non-trivial. For example, we can allow configurations with non-trivial winding $\int d\phi\,a_\phi = 2\pi k$. The configuration can be continued to the bulk smoothly while keeping $F = 0$ except for one singular point. For $k \in \mathbb{Z}$ this singular point is nothing but an invisible "Dirac point" (the 2d analogue of a Dirac string). We can extend the boundary conditions to the bulk by choosing the gauge $a_t = a_\phi$. Fixing the gauge and plugging $a_\mu = \partial_\mu \lambda$ into the action, one obtains

$$S = \frac{N}{4\pi}\int d\phi\,dt\left[\partial_t\lambda\partial_\phi\lambda - (\partial_\phi\lambda)^2\right]\,. \tag{21}$$

The result is that the theory is described by a chiral compact boson living on the boundary. Going back to our theory, we found that there is a chiral boson living on the ring. By coupling the theory to a background gauge field for the baryon symmetry, it can be shown that the baryon charge should be equivalent to the winding of the boson

$$B = \frac{1}{2\pi}\int d\phi\,\partial_\phi\lambda = \frac{1}{2\pi}\int d\phi\,a_\phi \,. \tag{22}$$

The configuration can be argued to be dynamically stable. There are various contributions to the energy of the configuration. Denote the radius of the ring by $R$. The potential for $\eta'$ contributes energy proportional to the area of the disc $\sim R^2$. The vanishing of the VEV of the chiral condensate on the ring contributes energy proportional to the perimeter of the ring $\sim R$. Finally, the edge mode contributes $\sim \frac{1}{R}$ due to its momentum on the ring. While the first two contributions want to minimize $R$, the last one prefers to increase it, resulting in some finite radius.

The spin of this configuration can be shown to be precisely $\frac{N}{2}$. The most convenient way to do it is in terms of the two-dimensional chiral theory living on the ring's worldsheet. The operator carrying one unit of baryon charge is the vertex operator $\mathcal{V}_N =: e^{iN\lambda}:$ whose spin is

$\frac{N}{2}$. Interestingly, in addition to $\mathcal{V}_N$, the theory contains also $\mathcal{V}_1 =: e^{i\lambda}:$ that carry fractional $\frac{1}{N}$ baryon charge. The appearance of this operator can be interpreted as having liberated quarks on the ring, that also carry $\frac{1}{N}$ baryon charge. See also [26] for a more elaborated discussion on this point. This is a summary of some of the main results of [5].

While this construction produces in a very non-trivial way many of the qualitative features of the $N_f = 1$ baryon, it raises some questions regarding the relation between this baryon and the skyrmion.

The two types of baryons are charged under two different symmetries. This is not what we expect to find. There should be one symmetry which is the low energy description of $U(1)_B$ and all the baryons should be charged under it. If, for example, we embed the $N_f = 1$ baryon inside $N_f = 2$ QCD, it should decay to a skyrmion, even though it carries no skyrmion charge, but some other topological charge.

Can these two charges be viewed as different descriptions of the same symmetry? Can we use this unified symmetry to understand the mechanism that allows $N_f = 1$ baryons to decay to skyrmions?

In the following sections we will try to answer these questions.

## 3 $N_f = 2 \rightarrow N_f = 1$ flow

In this section we will start from the hedgehog solution of the $N_f = 2$ chiral Lagrangian presented in section 2 and turn on a large mass for the second quark $m_d$. When doing so, we expect the mass difference between the skyrmion and the 1-flavored baryon to decrease, until at some point when the second quark is very massive, the 1-flavored baryon is expected to minimize the energy within the topological sector defined by $B = 1$. In the extreme limit where $m_d \rightarrow \infty$, the microscopic theory flows to $N_f = 1$ QCD and the 1-flavored baryon remains the only baryon in the spectrum. By including the $\eta'$ and continuously deforming the hedgehog to minimize the energy we will reproduce a very similar picture to the one constructed by Komargodski and described in section 2.2.

With the $\eta'$ included, we take the matrix $U \in U(2)$. We will parameterize the matrix as

$$U = e^{i\eta'/2}(\sigma + i\pi_a \tau_a), \ \sigma^2 + \pi_a^2 = 1 . \tag{23}$$

The matrix $U$ is invariant under

$$(\eta', \ \sigma, \ \pi_a) \rightarrow (\eta' + 2\pi, \ -\sigma, \ -\pi_a) . \tag{24}$$

For simplicity we will take for now the large $N$ limit where the $\eta'$ is massless and treat it as a NG boson, however nothing qualitative is expected to be different for finite $N$.

Our next step will be to add a mass term for the second quark. When the mass is small, the effect is to add to the chiral Lagrangian the following term

$$\mathcal{L}_M = tr(MU + MU^\dagger - 2M) = 2m_d(\cos(\eta'/2)\sigma + \sin(\eta'/2)\pi_3 - 1) , \tag{25}$$

where we took the mass matrix

$$M = \begin{pmatrix} 0 & 0 \\ 0 & m_d \end{pmatrix} . \tag{26}$$

As a result, three of the four NG bosons become massive. The mass term vanishes for $\pi_{1,2} = 0$ and $\sin(\eta'/2) = \pi_3$.

For a configuration with a non-trivial skyrmion charge, we cannot simply take all the massive fields to zero. It is obvious from the expression for the current (8) that we need the three

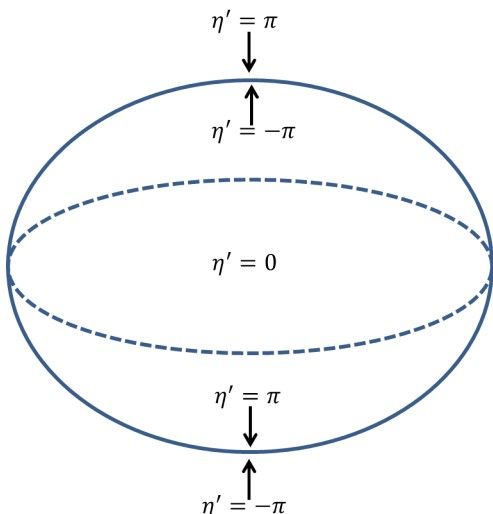

Figure 2: The value of $\eta' \in [-\pi, \pi]$ for the ansatz (28). The dashed line is the singular ring that connects the two $\eta' = \pi$ sheets. The value of $\eta'$ jumps by $\pm 2\pi$ as one crosses the sheets.

pions in order to get a non-trivial charge. For small mass, the hedgehog solution will be deformed in some small way to minimize the energy. If the mass of the down quark is very large, the solution will be highly deformed in a way that minimizes the volume in which the massive fields are non-zero. The first thing that we can do is to turn on a value for $\eta'$. $\eta'$ doesn't enter into the skyrmion current and we can use it to cancel at least some of the mass contribution. This is achieved by choosing

$$e^{i\eta'/2} = \frac{\sigma + i\pi_3}{\sqrt{\sigma^2 + \pi_3^2}} \ . \tag{27}$$

Notice that with this choice, the bottom-right entry of $U$ is exactly 1. Is this choice of $\eta'$ well defined? The denominator in (27) is zero when $\pi_3 = \sigma = 0$. Do such points exist in the skyrmion solution? For the hedgehog, it happens on the ring defined by $z = \cos(f) = 0$. Actually, this ring is a topological invariant in the sense that any topologically non-trivial mapping from $S^3$ to $S^3$ must include a ring on which $\sigma = \pi_3 = 0$. What about $\pi_{1,2}$? From the hedgehog solution, we see that $\pi_{1,2}$ are zero at $r \to \infty$ and on the z-axis. The regime in which they don't vanish has the shape of a bead, which can be continuously deformed to a ring, the same ring on which $\sigma = \pi_3 = 0$. We see that we can push all the massive fields to the ring, where outside the ring only the massless NG field is excited. We can suggest the following ansatz for the skyrmion solution in the large $m_d$ limit,

$$U_{ring} = e^{i\tilde{f}} \begin{pmatrix} e^{i\tilde{f}} \cos(h) & ie^{-i\phi} \sin(h) \\ ie^{i\phi} \sin(h) & e^{-i\tilde{f}} \cos(h) \end{pmatrix}, \tag{28}$$

where $\phi$ is as usual the angular coordinate along the ring, $h$ equals $\pi/2$ on the ring and goes to zero very fast outside of the ring, and $\tilde{f}$ winds once around the ring from 0 to $2\pi$.[5] (28) carries non-trivial topological charge $B = 1$, and it is a continuous deformation of the hedgehog solution. The behaviour of $\eta' = 2\tilde{f}$ is presented in figure 2.

---

[5]When continuously deforming the hedgehog to (28), it can be seen that $\tilde{f}$ is roughly $\text{sign}(z)f$. $f$ is even under $z \to -z$ and as you go around the ring, it varies from 0 to $\pi$ and back to 0 without any winding. $\tilde{f}$ on the other hand winds once from 0 to $2\pi$.

As we take $m_d \to \infty$, $h$ goes to 0 everywhere, such that (28) becomes

$$U_{ring} \to \begin{pmatrix} e^{2i\tilde{f}} & 0 \\ 0 & 1 \end{pmatrix} . \tag{29}$$

We see that as we flow to $N_f = 1$, the skyrmion transforms continuously to a configuration in which $\eta'$ winds around a singular ring, as in [5]. The winding of $\pi_{1,2}$ along the ring should be replaced by a winding of some new degree of freedom that appears on the singular ring. The construction of [5] tells us that this new degree of freedom is the chiral edge mode. In the next sections we will present the conservation law that ties these two windings together.

## 4 The Hidden symmetry

We will start this section by reviewing the conventional method for adding the vector mesons to the chiral Lagrangian using the idea of hidden gauge symmetry [6,7]. Next we will introduce a new "hidden" global symmetry and discuss its consequences.

The first step is to write the matrix $U$ in a redundant way as $U = \xi_L^\dagger \xi_R$ where $\xi_{L,R} \in U(2)$. The transformations

$$\xi_{L,R} \to h\xi_{L,R} , \ h \in U(2) , \tag{30}$$

are gauge transformations as the physical matrix $U$ is invariant under them. We can couple these transformations to dynamical gauge fields $V_\mu$ where as usual

$$D_\mu \xi_{L,R} = \partial_\mu \xi_{L,R} - iV_\mu \xi_{L,R} , \ V_\mu \to hV_\mu h^\dagger + ih\partial_\mu h^\dagger , \ F_{\mu\nu} = \partial_\mu V_\nu - \partial_\nu V_\mu - i[V_\mu, V_\nu] . \tag{31}$$

In addition, we also impose the $SU(N_f)_L \times SU(N_f)_R$ global symmetries :

$$\xi_L \to \xi_L g_L^\dagger , \ \xi_R \to \xi_R g_R^\dagger . \tag{32}$$

At the level of two derivatives we can write the following Lagrangian

$$\mathcal{L} = \frac{F_\pi^2}{4} \text{tr}(\partial_\mu(\xi_R^\dagger \xi_L)\partial^\mu(\xi_L^\dagger \xi_R)) - \frac{aF_\pi^2}{4} \text{tr}[D_\mu \xi_L \xi_L^\dagger + D_\mu \xi_R \xi_R^\dagger]^2 - \frac{1}{4g^2} F_{\mu\nu}^2 , \tag{33}$$

where $a$ is some dimensionless free parameter and $g$ is the coupling constant.

If we choose the unitary gauge $\xi_R = \xi_L^\dagger = \xi$ and $U = \xi^2$ we get

$$\mathcal{L} = \frac{F_\pi^2}{4} \text{tr}(\partial_\mu U^\dagger \partial^\mu U) - \frac{aF_\pi^2}{4} \text{tr}[\partial_\mu \xi \xi^\dagger + \partial_\mu \xi^\dagger \xi - 2iV_\mu]^2 - \frac{1}{4g^2} F_{\mu\nu}^2 , \tag{34}$$

which contains the usual kinetic terms for the pions and for the vector fields, a mass term for the vector fields and interactions between the vectors and the pions. Notice that even though we can expand $\xi$ locally in terms of the pions, we cannot write the interaction with the vector fields in terms of the original matrix $U$. Interestingly, using the "hidden" variables $\xi_{L,R}$ we can write a new skyrmion-like conserved current, we will denote by $H^\mu$. The construction is as follows. We can define the following currents from the $\xi_{L,R}$ matrices,

$$J_{L,R}^\mu = \frac{1}{24\pi^2} \left[ \epsilon^{\mu\nu\rho\sigma} \text{tr}(\xi_{L,R}^\dagger D_\nu \xi_{L,R} \xi_{L,R}^\dagger D_\rho \xi_{L,R} \xi_{L,R}^\dagger D_\sigma \xi_{L,R}) + \frac{3i}{2} \epsilon^{\mu\nu\rho\sigma} \text{tr}(F_{\nu\rho} D_\sigma \xi_{L,R} \xi_{L,R}^\dagger) \right] . \tag{35}$$

This form of currents, when replacing $\xi_{L,R} \to U$ is the correct version of the skyrmion current when coupled to chiral gauge fields [12,13,27]. The currents are manifestly gauge invariant, however they are not conserved. Instead

$$\partial_\mu J_{L,R}^\mu = \frac{1}{32\pi^2} \epsilon^{\mu\nu\rho\sigma} \text{tr}(F_{\mu\nu} F_{\rho\sigma}) . \tag{36}$$

An immediate result is that the current

$$H^\mu = J_R^\mu - J_L^\mu \,, \tag{37}$$

is manifestly gauge invariant and conserved identically. If we take the gauge $\xi_R = \xi_L^\dagger = \xi$, the current becomes

$$H^\mu = \frac{1}{24\pi^2} \epsilon^{\mu\nu\rho\sigma} \text{tr} \left[ 2\partial_\nu \xi \xi^\dagger \partial_\rho \xi \xi^\dagger \partial_\sigma \xi \xi^\dagger + 3i V_\nu (\partial_\rho \xi \partial_\sigma \xi^\dagger - \partial_\rho \xi^\dagger \partial_\sigma \xi) \right.$$
$$\left. + 3i \partial_\nu V_\rho (\partial_\sigma \xi \xi^\dagger - \partial_\sigma \xi^\dagger \xi) \right] \,. \tag{38}$$

At this point we should worry a little bit because it looks like there is an extra conserved current in the theory. We must understand how exactly it is related to the usual skyrmion current $B^\mu$. There are two possible logical scenarios. The first one is that the two currents $H^\mu$ and $B^\mu$ describe the same symmetry, i.e. every object charged under one, is also charged under the other. The second possibility is that the symmetries are different, and only one of them is exact and connected continuously to $U(1)_B$ of the uv theory.

In order to compare between the two symmetries, it is convenient to write $B^\mu$ in terms of $\xi$ using $U = \xi^2$. This results in

$$B^\mu = \frac{1}{24\pi^2} \epsilon^{\mu\nu\rho\sigma} \text{tr} \left[ 2\xi^\dagger \partial_\nu \xi \xi^\dagger \partial_\rho \xi \xi^\dagger \partial_\sigma \xi - 3\partial_\nu \xi \partial_\rho \xi \partial_\sigma (\xi^\dagger)^2 \right] \,. \tag{39}$$

We can see that the difference between the two currents is a full derivative,

$$H^\mu - B^\mu = \frac{1}{8\pi^2} \epsilon^{\mu\nu\rho\sigma} \partial_\sigma \text{tr} \left[ \partial_\nu \xi \partial_\rho \xi (\xi^\dagger)^2 + i V_\nu (\partial_\rho \xi \xi^\dagger - \partial_\rho \xi^\dagger \xi) \right] \,. \tag{40}$$

This means that assuming that everything is smooth and goes to the vacuum at infinity, the charges computed using each one of the currents will be the same. In particular, the hedgehog ansatz studied extensively in the literature is charged under $H^\mu$. The only difference between them is the local definition of current density. To emphasize this point, we can take the thousands of papers about skyrmions, and in all of them replace $B^\mu$ with $H^\mu$, and nothing bad will happen, they will still be correct. So it looks like the two currents describe the same symmetry. However, this is only true when dealing with smooth configurations in which the radius of the target space is finite everywhere. In principle, the definition and conservation of topological symmetries rely on the hidden assumption that the target space is well defined. If we are able to take the radius of the target space to zero somewhere, we can unwind the configuration and change the topological charge. This is true in general unless the topological symmetry is connected continuously to some symmetry in the uv. In this case, new degrees of freedom that carry the charge will appear on the singularities. From this perspective, it is clear that in order to distinguish between the two symmetries, we must study singular baryons, such as the $N_f = 1$ baryons. We will show next that the $N_f = 1$ baryon is charged under $H^\mu$ even though it is not charged under $B^\mu$. Therefore, we would like to suggest that $H^\mu$ is the correct description of the baryon current in the sense that it is connected to the $U(1)_B$ current in the uv.

The first interesting observation is that unlike $B^\mu$, $H^\mu$ is non-zero even when $N_f = 1$. The first two terms in (38) vanish because they involve anti-symmetrization over more than one generator, but the last term survives,

$$H^\mu(N_f = 1) = -\frac{1}{8\pi^2} \epsilon^{\mu\nu\rho\sigma} \partial_\nu \omega_\rho \partial_\sigma \eta' \,, \tag{41}$$

where we simply plugged into (38), $V_\mu = \omega_\mu$ , $\xi = e^{i\eta'/2}$. For later purposes, it will also be useful to derive this result by reduction of $N_f = 2$ to $N_f = 1$. For $N_f = 2$ we can parametrize

$$\xi = e^{i\eta'/4}(\alpha + i\beta_a \tau_a) \,, \ \alpha^2 + \beta_a^2 = 1 \,, \ \alpha^2 - \beta_a^2 = \sigma \,, \ 2\alpha\beta_a = \pi_a \,. \tag{42}$$

It is easy to verify that $\xi^2 = U$ as required. The only ambiguity in (42) is an overall sign $\xi \to -\xi$ which doesn't appear in any physical quantity. We will also denote the components of the vector meson by $V_\mu = \frac{1}{2}(\omega_\mu + \tau_a V_\mu^a)$. The last term in (38) is

$$\frac{i}{8\pi^2}\epsilon^{\mu\nu\rho\sigma}Tr[\partial_\mu V_\nu(\partial_\rho \xi\xi^\dagger - \partial_\rho \xi^\dagger \xi)] = -\frac{1}{16\pi^2}\epsilon^{\mu\nu\rho\sigma}Tr\left[\partial_\mu V_\nu\left(\partial_\rho \eta' + 4(\alpha\partial_\rho\beta_a - \beta_a\partial_\rho\alpha)\tau_a\right)\right]$$
$$= -\frac{1}{16\pi^2}\epsilon^{\mu\nu\rho\sigma}\left[\partial_\mu\omega_\nu\partial_\rho\eta' + 4\partial_\mu V_\nu^a(\alpha\partial_\rho\beta_a - \beta_a\partial_\rho\alpha)\right] \,. \tag{43}$$

When we reduce to $N_f = 1$, we should take

$$\pi_{1,2} = \beta_{1,2} = V_\mu^{1,2} = 0 \,, \ \omega_\mu = V_\mu^3 \,, \ e^{i\eta'/2} = \frac{\sigma + i\pi_3}{\sqrt{\sigma^2 + \pi_3^2}} \,. \tag{44}$$

The last equation is solved by

$$\pi_3 = \sin(\eta'/2) \,, \ \sigma = \cos(\eta'/2) \,, \ \alpha = \cos(\eta'/4) \,, \ \beta_3 = \sin(\eta'/4) \,. \tag{45}$$

Plugging it into the current, we again find (41). (41) is a non-trivial current that exists for $N_f = 1$. As stated above, the difference between $B^\mu$ and $H^\mu$ is a full derivative, and indeed (41) is also a full derivative. The integral over this current reduces to a boundary term at infinity, only if we can use Stokes theorem safely. However, this is not the case for our $N_f = 1$ baryon presented in figure 2. In order to compute its charge under (41), we will divide space into two regimes separated by the surface on which $|\eta'| = \pi$. In each one of the two regimes, $\eta'$ remains in its fundamental domain $\eta' \in [-\pi, \pi]$ and we can use Stokes theorem.

Recall that the value of $\eta'$ on the boundaries is:

| $\eta'$ on the boundaries: | Upper half of the surface | Lower half of the surface |
|---|---|---|
| From the outside | $\pi$ | $-\pi$ |
| From the inside | $-\pi$ | $\pi$ |

Therefore, the charge associated with $H^\mu$ is

$$H = -\frac{1}{8\pi^2}\epsilon^{ijk}\left[\int_{out} d^3x\partial_i\omega_j\partial_k\eta' + \int_{in} d^3x\partial_i\omega_j\partial_k\eta'\right]$$
$$= \frac{1}{4\pi}\epsilon^{ijk}\left[\int_{upper\ half} d^2x\hat{n}_k\partial_i\omega_j - \int_{lower\ half} d^2x\hat{n}_k\partial_i\omega_j\right] \,. \tag{46}$$

We can again integrate by parts and replace the two surface integrations with integral over the ring connecting the two surfaces. Parametrizing the coordinate on the ring as $\phi \in [0, 2\pi]$ we have

$$H = \frac{1}{2\pi}\int d\phi\,\omega_\phi \,. \tag{47}$$

(47) is identical to (22) if we identify the $\omega_\mu$ meson as the CS vector field living on the $\eta' = \pi$ domain wall. In the next sections we will explore this possibility.

Assuming for now that this is correct, we see that both the skyrmions and the $N_f = 1$ baryons are charged under the same current $H^\mu$. This construction gives a unified description for the two types of baryons. In the spirit of section 3, we can continuously deform the skyrmion to the singular $N_f = 1$ baryon. The winding of $\omega_\mu$ along the ring is inherited from the winding of $\pi_{1,2}$ along the ring as in (28). This point will be elaborated in 6.2.

## 5  $\omega_\mu$ as the Chern-Simons vector field

As was explained in 2.2, the CS domain wall theory plays an important role in the construction of the $N_f = 1$ baryon. In this section we will argue that the $\omega_\mu$ meson is actually the CS vector field on the domain wall. See also [28] for a related proposal. As part of this suggestion, we will propose to add to the Lagrangian the term

$$\mathscr{L}_{CS\eta'} = \frac{Ni}{8\pi^2}\epsilon^{\mu\nu\rho\sigma}\omega_\mu\partial_\nu\omega_\rho\, tr(\partial_\sigma\xi_R\xi_R^\dagger - \partial_\sigma\xi_L\xi_L^\dagger) = -\frac{N}{8\pi^2}\epsilon^{\mu\nu\rho\sigma}\omega_\mu\partial_\nu\omega_\rho\partial_\sigma\eta' \;. \tag{48}$$

The first evidence for the existence of (48) comes from the Wess-Zumino (WZ) term. As was shown in [29], under the gauging of a vectorlike U(1) global symmetry $U \to e^{iQ\alpha}Ue^{-iQ\alpha}$ where $Q$ is some diagonal matrix, gauge invariance of the WZ term requires adding to the theory[6]

$$\mathscr{L}_{GWZ} = NA_\mu J^\mu + \frac{iN}{24\pi^2}\epsilon^{\mu\nu\rho\sigma}\partial_\mu A_\nu A_\rho\, tr[Q^2\partial_\sigma UU^\dagger + Q^2U^\dagger\partial_\sigma U + QUQU^\dagger\partial_\sigma UU^\dagger] \;, \tag{49}$$

where $A_\mu$ is the associated gauge field and

$$J^\mu = \frac{1}{48\pi^2}\epsilon^{\mu\nu\rho\sigma} tr[Q\partial_\nu UU^\dagger\partial_\rho UU^\dagger\partial_\sigma UU^\dagger + QU^\dagger\partial_\nu UU^\dagger\partial_\rho UU^\dagger\partial_\sigma U] \;. \tag{50}$$

A surprising observation is that if we take $Q$ to be proportional to the identity, we get a nontrivial contribution even though the matrix $U$ is invariant under such transformation. In particular, by taking $Q = \frac{1}{N}$, we can recover the baryon current directly from (50)

$$Q = \frac{1}{N} \;\Rightarrow\; J^\mu = B^\mu \;. \tag{51}$$

We would like to make the following observation. From the hidden gauge principle, we know that $\omega_\mu$ is the U(1) gauge field of the transformation

$$\xi_{L,R} \to e^{i\lambda}\xi_{L,R} \;. \tag{52}$$

$U = \xi_L^\dagger\xi_R$ is of course gauge invariant, but following the same logic of [29], it is plausible to identify $\omega_\mu$ as the U(1) gauge field associated with taking $Q = 1$. With this identification, we find that the following terms should be added to the Lagrangian

$$\mathscr{L}_{top} = N\omega_\mu B^\mu + \frac{iN}{8\pi^2}\epsilon^{\mu\nu\rho\sigma}\partial_\mu\omega_\nu\omega_\rho\, tr[\partial_\sigma UU^\dagger] = N\omega_\mu B^\mu - \frac{N}{8\pi^2}\epsilon^{\mu\nu\rho\sigma}\omega_\mu\partial_\nu\omega_\rho\partial_\sigma\eta' \;. \tag{53}$$

Except for reproducing the desired term (48), we also notice that (53) can be written as $N\omega_\mu(H^\mu + ...)$ where ... stands for terms containing fields that do not exist in this construction (the traceless part of $V_\mu$ and $\xi_{L,R}$ in a combination that cannot be written in terms of $U$). It can be interesting to reproduce the entire coupling $N\omega_\mu H^\mu$ in a similar way. However, we leave this to future work.

---

[6]In [29] there is a minus sign in front of the first term in (49). The difference is due to different conventions for the covariant derivative and the gauge field transformation.

The addition of (48) to the Lagrangian has an interesting consequence. Consider the following domain wall configuration

$$\eta' = \eta'(z) \,, \quad \lim_{z\to-\infty} \eta'(z) = 0 \,, \quad \lim_{z\to\infty} \eta(z) = 2\pi \,. \tag{54}$$

We can ask what is the effective three dimensional theory living on the domain wall. At the classical level, this is done by expanding the fields around the background (54) and integrating over the $z$ direction. (48) then generates $\frac{N}{4\pi}\epsilon^{\mu\nu\rho}\omega_\mu\partial_\nu\omega_\rho$ which is exactly the $U(1)_N$ CS Lagrangian. In addition, from the other terms in (34) we get the usual Maxwell kinetic term (which is irrelevant in three dimensions) and a mass term for $\omega_\mu$. We conclude that the domain wall theory is a $U(1)_N$ Chern-Simons-Higgs (CSH) theory. Is this result consistent with our expectations? As explained above, the $\theta = \pi$ domain wall in YM must support a topological field theory such as $U(1)_N$ CS theory due to anomaly matching. In QCD, on the other hand, there is no 1-form symmetry and hence no 1-form anomaly. Therefore, it is reasonable to expect the domain wall theory to be a continuous deformation of $U(1)_N$ pure CS, where this deformation breaks the $\mathbb{Z}_N$ one-form symmetry without adding new light degrees of freedom. This expectation makes the CSH theory to be a very natural candidate for the domain wall theory (see also [22]). Indeed, thanks to (48), this theory is reproduced classically from the effective mesonic Lagrangian. One can also argue that when including (48), the $\eta'$ potential is actually generated by integrating out the $\omega_\mu$ meson.[7]

This is very similar to the effective model of [18]. In [18], it had been shown that the $\eta'$ potential can be generated from the VEV of the Gluonic topological density

$$Q = \frac{g^2}{64\pi^2}\epsilon^{\mu\nu\rho\sigma}G^a_{\mu\nu}G^a_{\rho\sigma} \,. \tag{55}$$

It happens in the following way. First, we fix the coupling between $Q$ and $\eta'$ such that shifts of $\eta'$ by a constant will generate a shift in the Lagrangian

$$\eta' \to \eta' + \alpha \;\Rightarrow\; \mathcal{L} \to \mathcal{L} - \alpha Q \,, \tag{56}$$

in accordance with the chiral anomaly. This is reproduced by the term

$$\mathcal{L}_{Q\eta'} = -\eta' Q \,, \tag{57}$$

where for simplicity we took the $\theta$ angle to be zero. In addition, we can write an effective theory for $Q$ that includes in the large $N$ limit only a quadratic term. The effective theory for $Q$ is given by the Lagrangian

$$\mathcal{L}_Q = \frac{1}{2F_\pi^2 M_{\eta'}^2}Q^2 - \eta' Q \,. \tag{58}$$

By integrating $Q$ out, we get[8]

$$\mathcal{L}_Q \to -\frac{F_\pi^2 M_{\eta'}^2}{2}\min_{k\in\mathbb{Z}}(\eta' + 2\pi k)^2 \,, \tag{59}$$

as in (13). It is also interesting to notice that the $\eta' = \pi$ domain wall theory can be read off directly from (57). As in the discussion after (54), it is straight forward to show that (57) generates an $SU(N)_{-1}$ CS term on the domain wall.

---

[7]We would like to thank Zohar Komargodski for pointing it out to us.

[8]When integrating $Q$ out, we should be careful about the periodicity of $\eta'$ and the quantization of $\int d^4 x Q \in \mathbb{Z}$. These lead to a periodic potential with a cusp at $\eta' = \pi$, instead of just a quadratic term.

The $SU(N)_{-1} \longleftrightarrow U(1)_N$ CS duality on the domain wall suggests a duality between the gluons and the vector mesons.[9] This is related to the conjecture that the vector mesons serve as Seiberg dual to the gluons [28, 30–32]. On the same way, a dual description for the source of the $\eta'$ potential involves integrating out the $\omega_\mu$ meson when (48) is present in the effective Lagrangian.[10]

The only ingredient left in the construction of the $N_f = 1$ baryon is understanding the edge modes living on the singular ring that serves as the boundary for the domain wall theory. To the best of our knowledge, edge modes in CSH theory with a boundary haven't been studied in the past. This issue will be discussed in the next section.

## 6 Edge modes quantization

### 6.1 $N_f = 1$

In this section we will discuss the existence of edge modes on the ring, giving rise to quantized value of the integral $\frac{1}{2\pi} \int d\phi \, \omega_\phi \in \mathbb{Z}$ as required in (47). The first thing we need to understand is what parts of the effective Lagrangian survive on the ring. For this we will add the so called dilaton field $\chi$ [33]. The conventional picture we are going to follow is (see for example [34, 35])

$$
\begin{aligned}
\mathscr{L}_{\eta'\omega\chi} = {} & \frac{1}{2}(\partial\chi)^2 + \frac{F_\pi^2}{4}\left(\frac{\chi}{F_\chi}\right)^2 (\partial_\mu\eta')^2 + \frac{aF_\pi^2}{4}\left(\frac{\chi}{F_\chi}\right)^2 (\partial_\mu S - 2\omega_\mu)^2 \\
& - \frac{1}{4g^2}F_{\mu\nu}^2 - \frac{N}{8\pi^2}\epsilon^{\mu\nu\rho\sigma}\omega_\mu\partial_\nu\omega_\rho\partial_\sigma\eta' - V_\chi ,
\end{aligned}
\tag{60}
$$

where $\partial_\mu S = -i(\partial_\mu\xi_L\xi_L^\dagger + \partial_\mu\xi_R\xi_R^\dagger)$ and $V_\chi$ is a potential that has a minimum at $\chi = F_\chi$. Its exact form will not be important for us. The appearance of $\chi$ to some power in front of the different terms is chosen to restore classical conformal symmetry where $\chi$ has scaling dimension 1. On the ring, $\partial\eta'$ is not well defined, and in order to get a finite energy configuration, $\chi$ must go to zero. The important point is that when this happens, the vector field $\omega_\mu$ becomes massless. The idea that the vector mesons become massless at high energies is an important ingredient of the Seiberg duality mentioned above. The fact that the topological term survives on the ring, even though $\eta'$ is not well defined seems a little bit problematic.[11] However, the consequence of this term on the ring is equivalent to having a Chern-Simons theory with a boundary. In particular, under gauge transformations $\omega_\mu \to \omega_\mu + \partial_\mu\lambda$ the action is no longer invariant, but

$$
\delta S_{\eta'\omega\chi} = -\frac{N}{8\pi^2}\int d^4x \, \epsilon^{\mu\nu\rho\sigma}\partial_\mu\lambda\partial_\nu\omega_\rho\partial_\sigma\eta' = -\frac{N}{2\pi}\int d^2x \, \lambda\epsilon^{ij}\partial_i\omega_j ,
\tag{61}
$$

where the last integral is over the ring's worldsheet and $i, j$ parametrize the coordinates on it. This problem can be solved by adding new physics on the boundary to restore gauge invariance. The simplest and minimal choice is to add a chiral boson. Assuming that such chiral

---

[9]This is a duality for pure CS theories. Since the domain wall theory is actually $U(1)_N$ CSH theory, the dual theory is expected to be $SU(N)_{-1}$ coupled to a fundamental fermion. It is not clear how the fermions enter into (58) since the theory is strongly coupled and uncontrolled. However, as in [22], we suggest that the domain wall theory contains also a fermion in a way consistent with the duality.

[10]Unlike [18], here we don't expect $\epsilon^{\mu\nu\rho\sigma}\partial_\mu\omega_\nu\partial_\rho\omega_\sigma$ to develop a VEV. It is more likely that the $\eta'$ potential comes from summing over $\omega_\mu$ instanton-like configurations.

[11]One might suggest that the naive power counting is not correct and that this term should also come with some powers of $\chi$ in front of it. However, gauge invariance implies that the coefficient $\frac{N}{8\pi^2}$ must be quantized and cannot flow continuously as we change the scale, similar to the level of a CS theory.

boson exists on the ring's worldsheet, we automatically reproduce all the results of [5]. It is interesting to understand the excitations of $\omega_\mu$ in the bulk and the similarities and differences from the massless case. If we didn't have a mass term for $\omega_\mu$ then the situation would have been similar to pure CS. The EOM $F = 0$ implies that $\omega_\mu$ should be locally a pure gauge everywhere. For example, $\omega_\mu = \delta_{\mu\phi}$. The singular string on the z-axis is nothing but an invisible Dirac string. The demand that the string should be invisible is equivalent to saying that $\frac{1}{2\pi} \int d\phi\, \omega_\phi \in \mathbb{Z}$.

What happens when the vector field is Higgsed? The EOM now don't force the field strength to vanish, and such configurations are not excluded from the spectrum. We can still have $F = 0$ everywhere but there is a price we need to pay. For $\omega_\mu = \partial_\mu \lambda$, we can excite $S$ to cancel the contribution from the mass term everywhere except for on the singular string. $S$ winds around the string and therefore on the string, $\chi$ must go to zero. Now the string is not a Dirac string anymore, but more similar to an abelian Higgs vortex. Instead of having an infinite string along the z-axis (which costs infinite amount of energy), we will have a vortex-loop circling the ring. To minimize the energy, the loop will shrink to infinitesimal radius until it is localized on the ring. It looks like the vortex will break the rotational symmetry $\phi \to \phi + c$. However, since the vortex becomes a local operator on the ring's worldsheet, it should be quantized as a two-dimensional excitation which cannot break continuous symmetries. In fact, it is tempting to interpret the vortex as the chiral boson living on the ring.

For all this procedure to work, we must:

- Forbid configurations with $\frac{1}{2\pi} \int d\phi\, \omega_\phi \notin \mathbb{Z}$: Such configurations will necessarily have non-zero magnetic field through the ring $F \neq 0$. They will cost more energy but we couldn't find any clear argument to exclude them from the spectrum.

- Forbid the vortex loop from crossing the ring or shrinking to zero size and disappearing completely.

The mechanism behind these two points might be related to the new physics that appear on the ring when $\chi \to 0$. We hope to gain better understanding of this in the future. At least for the second point, we can get some insights from gauge invariance. As we saw, the action (60) is gauge invariant in the presence of a ring only if the field strength on the worldsheet vanishes. This demand can be translated to the condition that

$$\partial_t \int d\phi\, \omega_\phi = 0 \Rightarrow \int d\phi\, \omega_\phi = const \,. \tag{62}$$

Therefore, any procedure that changes the winding of $\omega_\phi$ is forbidden. In particular, the vortex-loop circling the ring cannot cross the ring or decay completely. In the next section we will see that the same type of gauge invariance demand for $N_f = 2$ QCD connects $N_f = 1$ baryons with $N_f = 2$ skyrmions such that only the total baryon number is preserved.

## 6.2 Embedding in $N_f = 2$

In this section we will embed the $N_f = 1$ baryon inside the $N_f = 2$ theory, and study its decay to the regular skyrmion. The embedding is done by choosing a $U(1)$ subgroup inside $U(2)$ and taking all the other fields to zero

$$\pi_{1,2} = V_\mu^{1,2} = 0 \,, \quad \omega_\mu = V_\mu^3 \,, \quad e^{i\eta'/2} = \frac{\sigma + i\pi_3}{\sqrt{\sigma^2 + \pi_3^2}} \,, \quad V_\mu \equiv \frac{1}{2}(\omega_\mu + \tau_a V_\mu^a) \,. \tag{63}$$

An important ingredient that we add to the theory is the $N_f = 2$ completion of the topological term $\mathscr{L}_{CS\eta'}$ which we conjecture to be

$$\mathscr{L}_{top} = N \omega_\mu H^\mu \,, \tag{64}$$

where in the unitary gauge, can be written as

$$\mathcal{L}_{top} = \frac{N}{24\pi^2}\epsilon^{\mu\nu\rho\sigma}\omega_\mu \text{tr}\left[2\partial_\nu\xi\xi^\dagger\partial_\rho\xi\xi^\dagger\partial_\sigma\xi\xi^\dagger + 3iV_\nu(\partial_\rho\xi\partial_\sigma\xi^\dagger - \partial_\rho\xi^\dagger\partial_\sigma\xi) \right.$$
$$\left. + 3i\partial_\nu V_\rho(\partial_\sigma\xi\xi^\dagger - \partial_\sigma\xi^\dagger\xi)\right]. \tag{65}$$

We will use the parametrization (42). Assuming that $V_\mu^{1,2}$ will not play any role in the decay (at least qualitatively) we can set them identically to zero, such that the topological term becomes

$$\mathcal{L}_{top} = \frac{N}{12\pi^2}\epsilon^{\mu\nu\rho\sigma}\omega_\mu \text{tr}\left[\partial_\nu\xi\xi^\dagger\partial_\rho\xi\xi^\dagger\partial_\sigma\xi\xi^\dagger\right] + \frac{Ni}{8\pi^2}\epsilon^{\mu\nu\rho\sigma}\omega_\mu\partial_\nu\text{tr}\left[V_\rho(\partial_\sigma\xi\xi^\dagger - \partial_\sigma\xi^\dagger\xi)\right]$$
$$= \frac{N}{\pi^2}\epsilon^{\mu\nu\rho\sigma}\omega_\mu\left[(\beta_3\partial_\nu\alpha - \alpha\partial_\nu\beta_3)\partial_\rho\beta_1\partial_\sigma\beta_2 + \partial_\nu\alpha\partial_\rho\beta_3(\partial_\sigma\beta_1\beta_2 - \partial_\sigma\beta_2\beta_1)\right]$$
$$- \frac{N}{16\pi^2}\epsilon^{\mu\nu\rho\sigma}\omega_\mu\partial_\nu\left[\omega_\rho\partial_\sigma\eta' + 4V_\rho^3(\alpha\partial_\sigma\beta_3 - \beta_3\partial_\sigma\alpha)\right]. \tag{66}$$

Similar to the $N_f = 1$ case, we can compute the gauge variation of the action in the background of an $N_f = 1$ baryon. We need to be careful when integrating by parts due to the singular ring and several discontinuous fields. For the $N_f = 1$ baryon

$$\sigma = \cos(\eta'/2), \quad \pi_3 = \sin(\eta'/2) \Rightarrow \alpha\partial\beta_3 - \beta_3\partial\alpha = \frac{1}{4}\partial\eta'. \tag{67}$$

Therefore, the variation of the action is

$$\delta S_{top} = \frac{N}{4\pi^2}\int d^4x \epsilon^{\mu\nu\rho\sigma}\partial_\mu\lambda\left[-\partial_\nu\eta'\partial_\rho\beta_1\partial_\sigma\beta_2\right]$$
$$- \frac{N}{16\pi^2}\int d^4x \epsilon^{\mu\nu\rho\sigma}\partial_\mu\lambda\left[\partial_\nu\omega_\rho\partial_\sigma\eta' + \partial_\nu V_\rho^3\partial_\sigma\eta'\right]. \tag{68}$$

Integrating by parts carefully we get the worldsheet term

$$\delta S_{top} = -\frac{N}{4\pi}\int \epsilon^{ij}\lambda(\partial_i\omega_j + \partial_i V_j^3 + 4\partial_i\beta_1\partial_j\beta_2). \tag{69}$$

The gauge invariance condition is now modified such that only the combination

$$\epsilon^{ij}(\partial_i\omega_j + \partial_i V_j^3 + 4\partial_i\beta_1\partial_j\beta_2), \tag{70}$$

should vanish on the worldsheet. This can be translated to the constraint that

$$\int d\phi(\omega_\phi + V_\phi^3 + 4\beta_1\partial_\phi\beta_2) = const. \tag{71}$$

Starting from an $N_f = 1$ baryon with

$$\int d\phi(\omega_\phi + V_\phi^3) = 2\pi, \quad \beta_{1,2} = 0, \tag{72}$$

we find that it can decay to a configuration with

$$\omega_\phi + V_\phi^3 = 0, \quad \int d\phi\beta_1\partial_\phi\beta_2 = \frac{\pi}{2}. \tag{73}$$

Once the vector mesons are turned off, there is nothing that can prevent the ring with its $\eta'$ excitations from shrinking to zero radius and disappearing. When this happens we are left just with the pion fields excited. $\sigma$ and $\pi_3$ remain as they were before the vanishing of the ring. $\pi_{1,2}$ are excited such that on what was previously the ring

$$\int d\phi\, \beta_1 \partial_\phi \beta_2 = \frac{\pi}{2}\,. \tag{74}$$

Up to continuous deformations, this is exactly satisfied by (28). We already saw in section 3 how the $\eta'$ excitation is related to the $\pi_3$ and $\sigma$ excitations. This analysis shows the relation between the vector meson excitation to the $\pi_{1,2}$ excitations, completing the qualitative description of how the two different baryons can continuously transform one into the other.

### 6.3 A pancake or a pita?

In this section we would like to make a comment about the winding of $\eta'$. The $H = 1$ baryon (the minimal charge) in our setup looks different than the quantum Hall droplet discussed in [5]. The difference between the two is that the $\eta' = \pi$ surface in the quantum Hall droplet setup looks like a pancake, while in our setup it looks rather like a pita. Phrased more mathematically, in the setup of [5], there is one finite $\eta' = \pi$ surface that ends on the singular ring. Therefore, $\eta'$ winds once around the ring as can be seen in figure 1. On the other hand, in our setup there are two $\eta' = \pi$ surfaces stitched together on the singular ring, as illustrated in figure 2. This implies that in our setup $\eta'$ winds twice around the ring. If a pancake that carry edge modes indeed exists, it will have charge $\frac{1}{2}$ under $H^\mu$. Another way to view the problem with the pancake is that as we saw, when continuously flowing from $N_f = 2$ to $N_f = 1$, the skyrmion is deformed such that $\eta'$ winds twice around the ring. On the same way, if we try to embed the pancake in $N_f = 2$ QCD, (67) tells us that $\pi_3$ and $\sigma$ are not continuous even outside of the ring.

These arguments suggest that the pita is the minimal charged baryon, while the pancake should be excluded. However, these arguments come from $N_f = 2$, while the pancake looks perfectly fine in $N_f = 1$ QCD. In order to see what can go wrong with having a pancake from the $N_f = 1$ point of view, we will start by writing the fields $\xi_{L,R}$ before gauge fixing,

$$\xi_R = e^{\frac{i}{2}(S+\eta')}\,,\ \xi_L = e^{\frac{i}{2}(S-\eta')}\,. \tag{75}$$

Under $\eta' \to \eta' + 2\pi$, $\xi_{L,R} \to -\xi_{L,R}$. $-\xi_{L,R}$ is gauge equivalent to $\xi_{L,R}$ so the physics is indeed invariant under $\eta' \to \eta' + 2\pi$, but there are still consequences. To have a finite energy configuration, we must demand that $\xi_{L,R}$ are well defined as you go around the ring. It means that the sum of the windings of $\eta'$ and $S$ should be even. One possibility is that $S$ doesn't wind and $\eta'$ winds twice, which is exactly our pita. In the pancake, $\eta'$ winds once which means that also $S$ must wind once around the ring. While the two dishes are legitimate by themselves, we should ask whether they can be ordered with the extra special ingredient of edge modes. According to our analysis in 6.1, an edge mode requires $S$ to wind along the ring. As we just said, the pancake requires $S$ to wind around the ring. The two orthogonal windings of the same scalar $S$ hints that there is a physical difference between the pancake and the pita. Unfortunately, we don't have a good argument for excluding pancakes with edge modes, but we would like to suggest that they are excluded due to some (yet unknown) mechanism related to the windings of $S$, such that the pita is the minimal charge baryon.

## Acknowledgments

We would like to thank Pietro Benetti Genolini, Nick Dorey, Masazumi Honda, and Nick Manton for useful discussions. We would also like to especially thank Zohar Komargodski and David Tong for many discussions, insights and going over the draft, and the Blavatnik family foundation for the generous support. A.K is supported by the Blavatnik postdoctoral fellowship and partially by David Tong's Simons Investigator Award. This work has been also partially supported by STFC consolidated grant ST/P000681/1.

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
