# Peer review of "Skyrmions, Quantum Hall Droplets, and one current to rule them all"

_SciPost Physics, doi:SciPost Phys. 9, 008 (2020)_

## Round 2 · Referee Report · Anonymous (Referee 1) · 2020-4-7

Strengths

1) The paper nicely summarizes the existing constructions to describe baryons in terms of low energy descriptions in the literature.

2) The paper gives a nice unifying description that works for the two seemingly completely disconnected existing ideas for one flavor and two or more flavors.

3) The paper performs many checks to confirm the picture is correct.

Weaknesses

1) The paper is presumably the final word on the story, closing a loophole in our understanding of baryons as skyrmions. While it is very nice work, it is unlikely to stimulate large amounts of follow up work or open new doors.

Report

The low energy degrees of freedom of QCD are the pions, the Goldstone bosons of broken chiral symmetry. The correct low energy description of QCD is a theory of pions. Baryons in this language arise as non-perturbative objects made out of pions. At least this is the well established story for 2 or more flavors. For 1 flavor, the global symmetry is only U(1) baryon number to begin with and so there is no chiral symmetry breaking and hence no pions, so the story appears very different. At large N, a second U(1) appears, the otherwise anomalous axial U(1) whose anomaly is suppressed at large N, and so a new light boson appears as well, the eta'. It has been argued in previous work that in this case the baryon again can be written as some exotic soliton like excitation.

These two descriptions for the baryon both appeared individually correct, but are conceptually completely disconnected. This is puzzling, as the standard skyrmion for 2 or more flavors should go over to the new soliton in the limit that one makes all but one flavor very heavy. This is the problem solved in this paper. A unifying description is given, based on the hidden symmetry approach to QCD, that smoothly interpolates between the standard skyrmion description for 2 or more flavors and the novel soliton for 1 flavor.

---

## Round 3 · List of Changes

1) Section (6.3): Improved the explanation of the "pita" including a reference to a figure.
2) Typo corrected: Before equation 2.16 "U nder" -> "Under".
3) Before eq (2.18): Referred to a specific place for a detailed explanation about the Lorentz invariant boundary conditions

You are currently on this page

Resubmission 2003.07893v3 on 30 June 2020

---

## Editorial Decision

published